# Double burden of malnutrition and its associated factors among adolescents aged (10–19) years in a rural district of Pakistan

Muzna Hashmi[1], Naureen Rehman[1], Arjumand Rizvi[1], Zahra Ali Padhani[2],
Bhavita Kumari[1], Mushtaque Mirani[1], Muhammad Khan[1], Sana Khatoon[1],
Ayesha Zahid Khan[3], Rasool Bux[1], Imran Ahmed Chauhadry[4,5], Jai K. Das[1,4]*

**1** Institute for Global Health and Development, Aga Khan University, Karachi, Pakistan, **2** School of Public Health, Faculty of Health and Medical Sciences, University of Adelaide, Adelaide, Australia, **3** Human Development Program, Aga Khan University, Karachi, Pakistan, **4** Department of Pediatrics and Child Health, Aga Khan University, Karachi, Pakistan, **5** Centre of Excellence in Women and Child Health, Aga Khan University, Karachi, Pakistan

* jai.das@aku.edu

## Abstract

The double burden of malnutrition (DBM) is a growing concern in Pakistan, particularly among children and adolescents. This study assessed the prevalence and determinants of DBM in a rural district of Tando Muhammad Khan (TMK) in Pakistan. A representative cross-sectional study was conducted in a rural district of TMK using multistage cluster sampling. A total of 1,304 households were surveyed, and one eligible adolescent per household was selected. Anthropometric measurements were taken to assess nutritional status based on WHO criteria. Data were analyzed using multinomial logistic regression with survey design adjustments, and adjusted odds ratios (AORs) with 95% confidence intervals (CIs) were reported. Among 1,159 adolescents, the prevalence of underweight was 18.3% (95% CI: 15.0–20.0) and 4.2% were overweight/obese (95% CI: 3.0–5.0). The associated factors for underweight included working (AOR = 1.61, 95% CI: 1.01–2.50), no soap available/handwash supplies (AOR = 1.47, 95% CI: 0.99–2.20), inadequate sleep (AOR = 1.86, 95% CI: 1.34–2.59), and mild to moderate depression (AOR = 1.76, 95% CI: 1.04–3.00). The protective factors were being female (AOR = 0.57, 95% CI: 0.40–0.80) and always eating between meals (AOR = 0.64, 95% CI: 0.30–1.001). For overweight/obesity, snacking between meals was a risk factor (AOR = 2.00, 95% CI: 0.99–4.20), while larger family size was protective (AOR = 0.44, 95% CI: 0.24–0.80). Addressing DBM requires integrated strategies that improve hygiene, support mental health, and promote healthy eating. Gender-sensitive education and access to clean water, sanitation, and diverse diets are essential for better adolescent health outcomes.

**Data availability statement:** Data is provided in the supplementary file.

**Funding:** This study received internal institutional support from the Aga Khan University (AKU). No specific grant number was assigned for this work. The funding support was received by JKD. The funder provided institutional support only and had no role in the study design, data collection and analysis, decision to publish, or preparation of the manuscript. All authors receive(s) a salary from the Aga Khan University.

**Competing interests:** The authors have declared that no competing interests exist.

## Introduction

The global health landscape in low- and middle-income countries (LMICs) has significantly transformed, shifting from undernutrition to a "double burden of malnutrition" (DBM) [1]. This burden represents the coexistence of undernutrition (underweight) and overnutrition (overweight/obesity) within the same population groups, households, or communities. Initially identified in adults, this phenomenon now increasingly affects children and adolescents, raising serious public health concerns [2].

Adolescence is a crucial time for growth and development, yet much of the global research on malnutrition has been centered on children under the age of five years. According to the World Health Organization (WHO), adolescents aged 10–19 account for 20% of the global population, with 84% residing in LMICs [3]. Undernutrition in this age group impairs cognitive development, increases infection risk, and perpetuates intergenerational malnutrition. Concurrently, overnutrition contributes to early onset non communicable diseases (NCDs) like diabetes and cardiovascular diseases [4] Globally, adolescent obesity has quadrupled since 1975, with 18% of adolescents now overweight or obese, while 8.4% remain underweight [5].

South Asia faces a significant burden of malnutrition, in India, 24.4% of teenagers were thin and 4.8% were overweight, while in Bangladesh, underweight ranged from 12.7% to 16.3%, and overweight/obesity affected 10–15% [6,7]. In Pakistan, the situation is critical, with 21% of adolescent boys and 12% of girls underweight, and 18% of adolescent boys and 17% of girl's overweight or obese [8]. Furthermore, rural transformation, characterized by economic development, improved infrastructure, and changing lifestyles, is leading to increasing rates of overweight and obesity [9].

Recent research highlights various factors contributing to DBM among adolescents. Findings from India and Bangladesh indicate significant links between socioeconomic status (SES) and gender. Adolescents from lower SES backgrounds are more prone to undernutrition, whereas those from higher SES are at increased risk of obesity. Gender disparities have been documented, with higher stunting rates among girls in South Asia. Lifestyle factors, such as poor dietary diversity and insufficient physical activity, are also linked to DBM [10,11].

There's a notable gap in data about adolescents in Pakistan, especially those in rural areas. In contrast to previous studies that concentrated solely on school-going adolescents, our research encompasses both in-school and out-of-school individuals, thereby representing the full adolescent population. We address this gap by investigating adolescent malnutrition in Tando Muhammad Khan (TMK), an underexplored district in rural Pakistan. TMK represents a typical rural environment where economic disparities, dietary patterns, and limited healthcare access collectively influence nutritional outcomes. By studying adolescents across a range of socioeconomic backgrounds, educational statuses, and household conditions, this research explores how these factors contribute to the double burden of malnutrition. These insights are essential for developing targeted interventions and public health strategies to improve adolescent nutrition in similar low-resource settings.

## Materials and methods

### Ethics statement

Ethical approval was secured from the Aga Khan University's Ethical Review Committee (2021-5943-16892). Informed written consent from a legal guardian and assent from participants (10 to less than 18 years old) who agreed to participate in the study. Informed consent was taken from participants who are 18–19 years old. All participants were informed about the right to refuse or withdraw at any time from the survey without prejudice.

### Objectives

- To estimate the prevalence of underweight and overweight/obesity among adolescents aged 10–19 years in a rural district of Pakistan.
- To assess the sociodemographic, household, dietary, lifestyle, hygiene, and psychosocial factors associated with DBM (underweight, overweight/obesity) among adolescents aged 10–19 years in a rural district of Pakistan.

### Study setting

This study was conducted in the district of TMK in the province of Sindh in Pakistan, covering an area of 1,423 km² with a population of approximately 677,098, including 73,247 adolescents (2017 census) [12]. TMK is predominantly an agricultural area with 70% of the population working in farming. The district has 1,043 schools, mostly primary schools, but faces healthcare and sanitation challenges, with only 28% having improved sanitation [13].

### Study design and study population

This cross-sectional study was conducted to evaluate nutritional status among adolescents aged 10–19 years in TMK, Pakistan.

### Eligibility criteria

Adolescents aged 10–19 years, who were permanent residents of TMK were eligible for inclusion. Only one adolescent per household was selected, and participants enrolled in other trials were excluded.

In addition, adolescents who were pregnant, had a known psychiatric or cognitive condition impairing their ability to participate, or had a chronic medical condition were excluded.

### Sample size and sampling technique

To estimate the prevalence of underweight among adolescents aged 10–19 years with 95% confidence and 5% precision, the study determined a sample size based on a design effect of 2 and an anticipated response rate of 90% drawing from the National Nutrition Survey 2018, which reported underweight rates of 30.7% in boys and 20.1% in girls. For the sample size calculation, we used the higher prevalence of underweight reported for boys (30.7%) in the National Nutrition Survey 2018. This was selected to ensure an adequately powered sample for both sexes. A multistage cluster sampling method was used, selecting 48 rural clusters as primary sampling units from the CoMIC trial [14]. Systematic sampling was applied using a calculated interval (k) after a random starting point. If multiple adolescents in a household were eligible, the Kish grid method was used to select one participant. In cases of refusal, the next household in the random sequence was approached.

### Variables

The primary outcome was the "DBM" defined as underweight and overweight/obese individuals in the population. According to WHO criteria, underweight was defined as a weight-for-age less than -2 standard deviations (SD) from the WHO

Child Growth Standards median, while overweight/obese was defined as a weight-for-height greater than +1 standard deviation (SD) from the WHO Child Growth Standards median [15].

Anthropometric measurements of adolescents were conducted at the household level by trained field staff using standardized procedures. Weight and height were recorded to the nearest 0.1 kg and 0.1 cm, respectively, using calibrated Seca instruments (floor scale model 813 and stadiometer model 213), with participants wearing light clothing and no footwear. Each measurement was taken twice, and if discrepancies exceeded 0.5 kg for weight or 1 cm for height, a third measurement was performed by the team leader to ensure accuracy. Body Mass Index (BMI) was calculated as weight in kilograms divided by height in meters squared (kg/m²).

## Covariates and their measurement tools

To examine determinants of the double burden of malnutrition among adolescents, the following validated tools and scales were used:

Sociodemographic, Household, and Nutritional Characteristics

- Socioeconomic status: Assessed using items adapted from the Pakistan Demographic and Health Survey (PDHS) [16].

- Dietary diversity: Measured using the FANTA Household Dietary Diversity Scale (HDDS) [17].

- Household food insecurity: Assessed using the Global Food Insecurity Experience Scale (FIES) [18].

Lifestyle, Dietary, Hygiene, and Psychosocial Factors

- Lifestyle factors (meal patterns, physical activity, dental hygiene, sleep quality, general health): Assessed using tools from the WHO School-based Health Survey [19].

- Dietary intake (macronutrients and micronutrients): Assessed using a combination of a 24-hour dietary recall and a semiquantitative food frequency questionnaire. Food composition data were sourced from the MAL-ED Pakistan database, USDA National Nutrient Database, and Food Composition Table for Bangladesh, with retention factors applied for cooking losses [20–30].

- Water, Sanitation and Hygiene (WASH) practices: Evaluated using PDHS questions [16].

- Psychosocial factors (child labor, child discipline, tobacco use): Assessed using items from the Multiple Indicator Cluster Survey (MICS) [31].

- Depression: Measured using the Beck Depression Inventory (BDI) [32].

## Data collection

The recruitment of study participants was conducted from 01/03/2021–31/05/2021. A two-staged sampling technique was used; the sample size was equally divided between clusters and households were selected for a calculated interval (k) after a random starting point. The household information, including the demographic and socio-economic measures, and health outcomes were obtained from the mother or caregiver after obtaining informed consent. Data were collected by trained data collectors using handheld devices (Samsung tablets running Android 5.1) after 6 days of training on content, operational procedures, and management. Range and consistency checks and skip patterns were built into the data entry program to minimize errors. Data were synced daily and uploaded from the field sites to the university server. The data management unit generated daily summary reports for quality check and, if required, sent the reports to the field teams for rectification. All the data were encrypted, secured, and fully anonymized.

## Statistical analysis

All analyses were conducted using Stata 17, applying survey weights to account for the complex sampling design. Missing data was minimal (<1%). Pairwise diagnostics showed no systematic association between missingness and observed variables, supporting the use of complete-case analysis. Descriptive statistics (frequencies and percentages) summarized sociodemographic variables, and cross-tabulations explored their distribution across nutritional outcomes. Multinomial logistic regression was used for model building. Univariate analyses identified potential predictors of the double burden of malnutrition, with variables significant at p<0.25 included in the multivariable model. Multicollinearity among categorical variables was assessed using Cramer's V. Results were reported as adjusted odds ratios (AORs) with 95% confidence intervals.

Dietary intake was assessed using both a semiquantitative food frequency questionnaire and 24-hour dietary recall. Nutrient intake was calculated by multiplying the frequency weight, portion size, and nutrient content for each food item. Food composition data were sourced from the MAL-ED Pakistan database and supplemented with international references when necessary. A trained nutritionist assigned food codes and applied retention factors to account for nutrient losses due to cooking. Total energy, macronutrient, and micronutrient intakes were computed for each participant and incorporated into the regression models. The data used for this study is available as a supplementary file (S1 Data).

## Results

Of the 1,304 households selected through systematic sampling, 119 did not have an eligible adolescent because the house was locked (n=63), refused participation (n=31), or did not have an adolescent aged 10–19 years (n=25). After these exclusions, 1,185 households had one eligible adolescent, corresponding to 1,185 eligible adolescents. Subsequently, 26 adolescents were further excluded due to pregnancy (n=8), psychiatric conditions (n=3), or chronic diseases (n=15), leaving a final sample of 1,159 adolescents (565 boys, 594 girls) (Fig 1).

### Descriptive analysis

Of the 1,159 adolescents included in the study, 55.2% were 10–14 years and 44.7% were 15–19 years old. Nearly half were males (48.7%), and most participants identified as Muslims (75.9%). Most households had more than five members (64.4%), and 80.9% shared a single room for sleeping. Sanitation conditions were suboptimal, with 70.2% using unimproved facilities, although 93.2% had water available at handwashing stations. A significant portion of the parents were uneducated: 55.3% of fathers and 87.4% of mothers. Additionally, 48.2% of adolescents were uneducated, and only 5.4% of the adolescents were married at the time of the survey (Table 1).

### Double burden of malnutrition

Among adolescents in the study, 18.3% were underweight (95% CI: 15.0% – 20.0%) and 4.2% were overweight or obese (95% CI: 3.0% – 5.0%). The coexistence of both undernutrition and overnutrition within this population demonstrates the double burden of malnutrition in rural TMK (Table 1).

**Double burden of malnutrition among adolescents by demographic characteristics.** In univariate analysis, age, gender, family size, socioeconomic status, sanitation facilities, water availability for handwashing, mother's education, and food insecurity met the inclusion threshold (p<0.25) and were considered for multivariable analysis. Adolescents aged 13–15 years had reduced odds of underweight (COR: 0.7; 95% CI: 0.5–1.1) and overweight/obesity (COR: 0.5; 95% CI: 0.2–1.2) compared to those aged 10–12 years. Females were significantly less likely to be underweight than males (COR: 0.6; 95% CI: 0.4–0.9). Larger households (>5 members) were associated with higher odds of underweight (COR: 1.3; 95% CI: 0.9–2.0) and lower odds of overweight/obesity (COR: 0.4; 95% CI: 0.2–0.8). Limited water access for handwashing showed increased odds of underweight (COR: 1.5; 95% CI: 0.8–2.6) and reduced odds of overweight/

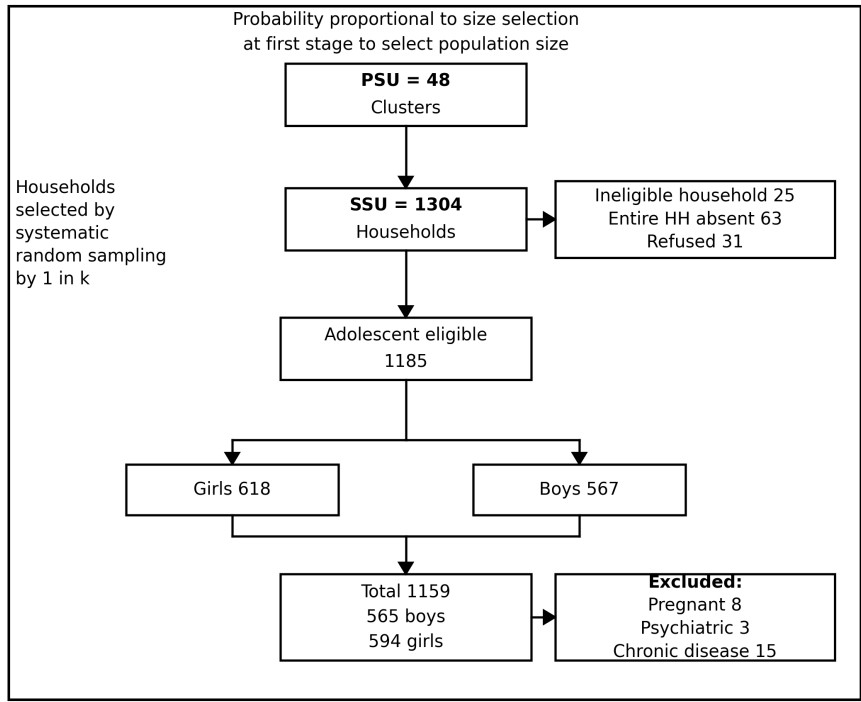

**Fig 1. Sampling and participant selection flowchart.**

obesity (COR: 0.25; 95% CI: 0.03–2.0). Severe food insecurity was also associated with increased odds of underweight (COR: 1.4; 95% CI: 0.8–2.5) (Table 2).

**Double burden of malnutrition among adolescents in relation to lifestyle, dietary, hygiene, and psychosocial factors.** In univariate analysis, physical activity, sleep quality in the past week, eating between meals, intake of vitamin B6, vitamin A, zinc, protein, availability of handwashing materials, and working hours met the inclusion threshold ($p < 0.25$) and were considered for multivariable analysis. Physical inactivity (<180 min/week) was associated with higher odds of being overweight/obese (COR 2.2, 95% CI: 0.6–8.4). Adolescents reporting average sleep quality had increased odds of being underweight (COR 1.8, 95% CI: 1.3–2.4), whereas those with poor sleep had slightly higher overweight/obese prevalence (6.1%). Eating between meals almost every day was associated with greater odds of overweight/obese (COR 1.6, 95% CI: 0.8–3.2).

Inadequate vitamin B6 intake was associated with higher odds of underweight (COR 1.3, 95% CI: 0.9–1.8) and lower odds of overweight/obese (COR 0.51, 95% CI: 0.2–0.9). Zinc inadequacy showed similar trends with increased likelihood of underweight (COR 2.0, 95% CI: 1.2–3.2) and reduced likelihood of overweight/obese (COR 0.5, 95% CI: 0.2–1.1). Adolescents with inadequate energy intake had higher odds of being underweight (COR 2.19, 95% CI: 0.7–6.2).

Adolescents reporting no handwashing materials available had higher odds of underweight (COR 1.3, 95% CI: 0.9–2.0) and lower odds of overweight/obese (COR 0.4, 95% CI: 0.1–1.2), relative to those who used bar soap.

Among psychosocial factors, adolescents working 0–3 hours per day had higher odds of being underweight (COR 1.8, 95% CI: 1.2–2.7), while mild to moderate depression was also associated with greater odds of underweight (COR 1.7, 95% CI: 1.06–2.9). No significant associations were found between child labor status and overweight/obese (Table 3).

**Table 1. Weighted percentages of sociodemographic, household, and nutritional characteristics of study participants (N = 1,159).**

| Variable | Total N = 1159 |
|---|---|
| **Age group** | |
| 10-12 | 360(31.7) |
| 13-15 | 421(36.3) |
| 16-19 | 378(32.6) |
| **Gender** | |
| Male | 565(48.7) |
| Female | 594(51.3) |
| **Family size** | |
| >5 | 746(64.4) |
| **Sanitation facilities** | |
| Improved | 326(28.1) |
| Unimproved | 814(70.2) |
| **Water presence at handwashing** | |
| Water available | 1070(92.3) |
| Water not available | 89(7.7) |
| **Father education level** | |
| Primary | 255(22.2) |
| Secondary | 144(12.5) |
| College and higher | 112(9.7) |
| Uneducated | 633(55.3) |
| **Mother education level** | |
| Primary | 104(8.9) |
| Secondary | 20(1.7) |
| College and higher | 6(0.5) |
| Uneducated | 1014(87.4) |
| **Current education status of adolescent** | |
| School going | 429(36.9) |
| Out of school | 727(63.1) |
| **Marital status of adolescents** | |
| Married | 64(5.3) |
| Unmarried | 1094(94.6) |
| **Nutritional Characteristics** | |
| **Food insecurity** | |
| Food secure | 159(13.7) |
| Mild food secure | 104(8.9) |
| Moderate food secure | 130(11.2) |
| Severely unsecure | 765(66.0) |
| **Dietary Diversity** | |
| Lowest dietary diversity | 84(7.2) |
| Medium dietary diversity | 767(66.2) |
| High dietary diversity | 308(26.5) |
| **BMI** | |
| Underweight ≤ -2SD | 213(18.3) |
| Normal -2SD- + 1SD | 897(77.3) |
| Overweight/obese >+1SD | 49(4.2) |

**Table 2. Weighted percentages of double burden of malnutrition among adolescents by demographic characteristics.**

| Variable | Total N = 1159 (n) | Normal weight | Under weight | Over weight/ obese | COR underweight | COR Overweight/obese |
|---|---|---|---|---|---|---|
| **Age group** | | | | | | |
| 10-12 | 360 | 75.2 | 19.7 | 5.1 | Ref | Ref |
| 13-15 | 421 | 80.5 | 16.3 | 3.2 | 0.7(0.5-1.1) * | 0.5(0.2-1.2) * |
| 16-19 | 378 | 76.6 | 18.8 | 4.6 | 0.9(0.6-1.4) | 0.8(0.4-1.9) |
| **Gender** | | | | | | |
| Male | 565 | 74.1 | 21.4 | 4.5 | Ref | Ref |
| Female | 594 | 81.0 | 15.2 | 3.8 | 0.6(0.4-0.9) * | 0.7(0.4-1.4) |
| **Socioeconomic status** | | | | | | |
| High | 469 | 75.8 | 19.9 | 4.3 | Ref | Ref |
| Middle | 232 | 80.3 | 16.7 | 3.0 | 0.7(0.5-1.1) * | 0.6(0.2-1.7) |
| Low | 458 | 78.5 | 17.0 | 4.5 | 0.8(0.5-1.1) | 0.9(0.5-1.9) |
| **Family size** | | | | | | |
| <=5 members | 413 | 79.0 | 14.6 | 6.4 | Ref | Ref |
| >5 members | 746 | 77.0 | 20.0 | 3.0 | 1.3(0.9-2.0) * | 0.4(0.2-0.8) * |
| **Sanitation facilities** | | | | | | |
| Unimproved | 814 | 76.7 | 19.2 | 4.1 | 1.3(0.9-1.9) * | 1.0(0.5-1.9) |
| Improved | 326 | 80.5 | 15.2 | 4.3 | Ref | Ref |
| **Water presence at handwashing** | | | | | | |
| Water available | 1070 | 78.0 | 17.6 | 4.4 | Ref | Ref |
| Water not available | 89 | 73.3 | 25.7 | 1.0 | 1.5(0.8-2.6) * | 0.2(0.03-2.0) * |
| **Father education** | | | | | | |
| Uneducated | 633 | 77.8 | 18.2 | 3.7 | Ref | Ref |
| Primary | 255 | 76.9 | 18.4 | 3.7 | 1.0(0.6-1.4) | 0.9(0.4-2.0) |
| Secondary | 144 | 73.7 | 20.7 | 4.8 | 1.2(0.7-1.8) | 1.3(0.4-3.4) |
| College and Higher | 112 | 81.5 | 12.5 | 6.0 | 0.64(0.3-1.1) | 1.4(0.5-3.7) |
| **Mother education** | | | | | | |
| Uneducated | 1014 | 77.8 | 18.1 | 4.1 | Ref | Ref |
| Primary | 104 | 79.0 | 17.6 | 3.4 | 1.3(0.8-1.9) | 0.7(0.3-1.2) |
| Secondary | 20 | 74.9 | 21.0 | 4.1 | 1.3(0.8-2.0) * | 1.1(0.5-2.4) |
| College and higher | 6 | 53.8 | 22.9 | 23.3 | 1.2(0.4-3.1) | 0.5(0.06-5.2) |
| **Ever attended school adolescents** | | | | | | |
| School going | 601 | 76.7 | 19.4 | 3.9 | Ref | Ref |
| Out of school | 542 | 78.7 | 16.9 | 4.4 | 0.84(0.6-1.1) | 1.1(0.5-2.1) |
| **Food insecurity** | | | | | | |
| Food secure | 159 | 82.2 | 14.4 | 3.4 | Ref | Ref |
| Mild food secure | 104 | 79.9 | 14.3 | 5.8 | 1.0(0.4-2.1) | 1.7(0.3-7.4) |
| Moderate food secure | 130 | 79.4 | 17.7 | 2.9 | 1.2(0.6-2.6) | 0.8(0.2-3.7) |
| Severely unsecure | 765 | 76.2 | 19.5 | 4.3 | 1.4(0.8-2.5) * | 1.3(0.4-4.1) |
| **Household dietary diversity** | | | | | | |
| High dietary diversity(≥6 food groups) | 308 | 77.3 | 18.5 | 4.2 | Ref | Ref |
| Lowest dietary diversity (≤ 3 food group) | 84 | 75.6 | 20.5 | 3.9 | 1.1(0.6-2.0) | 0.9(0.2-3.5) |
| Medium dietary diversity (4 and 5 food group) | 767 | 78.1 | 17.6 | 4.3 | 0.9(0.6-1.3) | 0.9(0.5-1.8) |

**\*p<0.25**.

**Table 3. Weighted percentages of double burden of malnutrition among adolescents in relation to lifestyle, dietary, hygiene, and psychosocial factors (N = 1159).**

| | Total N = 1159 (n) | Normal weight | Under weight | Over weight/ obese | COR underweight | COR Overweight/obese |
|---|---|---|---|---|---|---|
| **Lifestyle factors** | | | | | | |
| **Physical activity** | | | | | | |
| Vigorous Physical Activity(above 300) | 141 | 80.1 | 17.8 | 2.1 | Ref | Ref |
| Inactive (<180 min/week) | 901 | 77.7 | 17.7 | 4.6 | 1.0(0.6-1.6) | 2.2(0.6-8.4)* |
| Moderate-Vigorous (180–300min/week) | 117 | 74.3 | 22.0 | 3.7 | 1.3(0.7-2.5) | 1.8(0.3-10.7) |
| **Sleep quality in past week** | | | | | | |
| Good | 824 | 80 | 15.9 | 4.1 | Ref | Ref |
| Average | 258 | 70.5 | 25.6 | 3.9 | 1.8(1.3-2.4) * | 1.0(0.4-2.5) |
| Poor | 64 | 83.9 | 10.0 | 6.1 | 1.2(0.5-2.6) | 1.5(0.5-4.6) |
| **Dietary Intake: Meal Patterns and Micro/Macro Nutrients** | | | | | | |
| **Eating between meals(snacking)** | | | | | | |
| Never | 403 | 76.4 | 20.3 | 3.3 | Ref | Ref |
| Sometimes | 483 | 78.1 | 18.0 | 3.9 | 0.8(0.6-1.2) | 1.1(0.5-2.2) |
| Almost every day | 273 | 79 | 15.3 | 5.7 | 0.7(0.4-1.1) * | 1.6(0.8-3.2) * |
| **Micronutrient (Thiamin, Riboflavin, Niacin, Vitamins A, B6, C, D, E, Folate, Calcium, Zinc, and Iron)** | | | | | | |
| Adequate | 524 | 20.5 | 76.3 | 3.07 | Ref | Ref |
| Inadequate | 631 | 16.8 | 78.9 | 4.17 | 0.79(0.5-1.0) | 1.3 (0.7-2.4) |
| **Vitamin B6** | | | | | | |
| Adequate | 415 | 15.6 | 78.9 | 5.39 | Ref | Ref |
| Inadequate | 740 | 20.1 | 77.1 | 2.72 | 1.3(0.9-1.8)* | 0t.51(0.20.9)* |
| **Vitamin A** | | | | | | |
| Adequate | 16 | 12.5 | 81.2 | 6.25 | Ref | Ref |
| Inadequate | 1139 | 18.6 | 77.7 | 3.63 | 1.5(0.3-6.9) | 0.6(0.07-4.7) |
| **Vitamin D** | | | | | | |
| Adequate | 11 | 9.09 | 81.8 | 9.09 | Ref | Ref |
| Inadequate | 1144 | 18.6 | 77.7 | 3.62 | 2.16(0.21-7.1) | 0.4(0.05-3.3) |
| **Zinc** | | | | | | |
| Adequate | 196 | 10.77 | 83.08 | 6.15 | Ref | Ref |
| Inadequate | 959 | 20.13 | 76.71 | 3.16 | 2.0(1.2-3.2)* | 0.5(0.2-1.1)* |
| **Iron** | | | | | | |
| Adequate | 382 | 17.51 | 78.51 | 3.98 | Ref | Ref |
| Inadequate | 773 | 19.04 | 77.44 | 3.52 | 1.10(0.8-1.5) | 0.89(0.4-1.7) |
| **Carbohydrate** | | | | | | |
| Adequate | 1141 | 18.5 | 77.79 | 3.72 | Ref | Ref |
| Inadequate | 14 | 21.43 | 78.57 | 0 | 1.15(0.3–4.1) | – |
| **Protein** | | | | | | |
| Adequate | 763 | 21.7 | 75.2 | 3.03 | Ref | Ref |
| Inadequate | 389 | 12.3 | 83.5 | 4.1 | 0.5(0.3-0.7)* | 1.2(0.6-2.3) |
| **Fat** | | | | | | |
| Adequate | 821 | 18.55 | 77.89 | 3.56 | Ref | Ref |
| Inadequate | 334 | 18.48 | 77.58 | 3.94 | 1.0(0.7–1.3) | 1.11(0.5–2.1) |
| **Fiber** | | | | | | |
| Adequate | 198 | 14.87 | 80 | 5.13 | Ref | Ref |

*(Continued)*

**Table 3.** (Continued)

| | Total N = 1159 (n) | Normal weight | Under weight | Over weight/ obese | COR underweight | COR Overweight/obese |
|---|---|---|---|---|---|---|
| Inadequate | 957 | 19.28 | 77.34 | 3.37 | 1.34(0.8–2.0) | 0.68(0.3–1.4) |
| **Energy** | | | | | | |
| Adequate | 41 | 9.76 | 87.80 | 2.44 | Ref | Ref |
| Inadequate | 1114 | 18.86 | 77.43 | 3.72 | 2.19(0.7–6.2) | 1.7(0.2–12.9) |
| **Hygiene** | | | | | | |
| **Hand wash materials** | | | | | | |
| Bar Soap | 754 | 77.7 | 17.3 | 5.0 | Ref | Ref |
| Not Available | 283 | 74.9 | 23.0 | 2.1 | 1.3(0.9-2.0)* | 0.4(0.1-1.2)* |
| Ash/Mud | 68 | 88 | 9.0 | 3.0 | 0.4(0.2-1.0)* | 0.5(0.1-2.4) |
| Detergent/liquid soap | 39 | 81.5 | 12.16 | 6.3 | 0.6(0.2-1.6) | 1.2(0.3-4.5) |
| **Social Factors** | | | | | | |
| **Child Labor** | | | | | | |
| Not Working | 750 | 78.4 | 17.4 | 4.2 | Ref | Ref |
| Working | 409 | 76.4 | 19.6 | 4.0 | 1.1(0.8-1.6) | 0.9(0.5-1.8) |
| **Working Hours** | | | | | | |
| Not working | 751 | 78.4 | 17.3 | 4.3 | Ref | Ref |
| 0-3 hours | 130 | 68 | 28.1 | 3.9 | 1.8(1.2-2.7)* | 1.0(0.4-2.6) |
| 4-6 hours | 179 | 82 | 14.6 | 3.4 | 0.8(0.4-1.3) | 0.7(0.2-1.9) |
| 7-12 hours | 99 | 77 | 17.9 | 5.1 | 1.0(0.6-1.8) | 1.1(0.4-3.2) |
| **Depression** | | | | | | |
| Minimal or no depression (1–10) | 1035 | 78.54 | 17.26 | 4.2 | Ref | Ref |
| Mild to moderate depression (>11) | 109 | 69.69 | 26.31 | 4.0 | 1.7 (1.06-2.9) | 1.17(0.4-3.2) |

**\*p<0.25**.

**Multivariable analysis.** The analysis revealed several associations with DBM. For being underweight, we identified several associated factors; adolescents working 0–3 hours per day had higher odds of being underweight (AOR: 1.61; 95% CI: 1.01–2.5), not having soap for hand washing also increased the odds of being underweight (AOR: 1.47; 95% CI: 0.99–2.20), average sleep quality was also associated with underweight (AOR: 1.86; 95% CI: 1.34–2.59), and those with mild to moderate depression were more likely to be underweight (AOR: 1.76; 95% CI: 1.04–3.00). On the other hand, being female was protective against being underweight (AOR: 0.57; 95% CI: 0.40–0.81), as was always snacking between meals (AOR: 0.64; 95% CI: 0.39–1.001).

For being overweight/obese; always eating between meals was identified as an associated factor (AOR: 2.00; 95% CI: 0.95–4.24). However, coming from a larger family seemed to protect against being overweight/obese (AOR: 0.44; 95% CI: 0.24–0.80) (Table 4).

## Discussion

This study examined the prevalence and associated factors of DBM among adolescents in rural Pakistan, suggesting 18.3% underweight and 4.2% overweight or obese.

Globally, the prevalence of DBM varies significantly by region. A meta-analysis of 62,148 children aged 5–15 in Pakistan reported an undernutrition prevalence of 25.1% and combined overweight/obese rates of 11.4% and 6.9%,

**Table 4. Multinomial logistic regression model for factors associated with double burden of malnutrition.**

| Variable | aCOR (95% bCI) for underweight | cAOR (95% CI) for underweight | COR(95% CI) for overweight/obese | AOR (95% CI) for overweight/obese |
|---|---|---|---|---|
| **Gender** | | | | |
| Male (Ref) | Ref | Ref | Ref | Ref |
| Female | 0.6 (0.4-0.9) | 0.5 (0.4-0.8)* | 0.76(0.4-1.4) | 0.8(0.4-1.5) |
| **Working Hours** | | | | |
| Not Working (Ref) | Ref | Ref | Ref | Ref |
| 0-3 Hours | 1.86 (1.25-2.79) | 1.61(1.01-2.5)* | 1.05 (0.4-2.6) | 1.07 (0.4-2.6) |
| 4-6 Hours | 0.80 (0.49-1.32) | 0.6(0.4-1.1) | 0.7(0.3-1.9) | 0.7(0.2-1.9) |
| 7-12 Hours | 1.05 (0.60-1.82) | 0.7 (0.4-1.3) | 1.2(0.4-3.2) | 1.05(0.3-3.3) |
| **Eating Between Meals(snacking)** | | | | |
| Never | Ref | Ref | Ref | Ref |
| Sometimes | 0.87 (0.61-1.23) | 0.7 (0.5-1.07) | 1.1 (0.5-2.2) | 1.44 (0.6-3.01) |
| Always | 0.73 (0.4-1.1) | 0.6 (0.3-1.001)* | 1.6(0.8-3.2) | 2.0 (0.99-4.2)* |
| **Hand wash supplies** | | | | |
| Bar Soap Available | Ref | Ref | Ref | Ref |
| No Soap Available | 1.37 (0.94-2.01) | 1.47 (0.99-2.20)* | 0.45 (0.17-1.23) | 0.40 (0.14-1.17) |
| Ash/Mud | 0.46 (0.20-1.03) | 0.53 (0.22-1.3) | 0.53 (0.12-2.42) | 0.45 (0.09-2.23) |
| Detergent/liquid | 0.66 (0.26-1.70) | 0.62 (0.23-1.69) | 1.21 (0.32-4.52) | 0.67 (0.13-3.39) |
| **Sleep Quality** | | | | |
| Good | Ref | Ref | Ref | Ref |
| Average | 1.83 (1.35-2.48) | 1.86 (1.34-2.59)* | 1.08 (0.47-2.52) | 0.94 (0.38-2.30) |
| Poor | 1.20 (0.55-2.64) | 1.01 (0.45-2.21) | 1.60 (0.55-4.60) | 1.75 (0.60-5.15) |
| **Family Size** | | | | |
| <5 Members | Ref | Ref | Ref | Ref |
| >5 Members | 1.39 (0.93-2.08) | 1.40 (0.93-2.10) | 0.46 (0.25-0.84) | 0.44 (0.24-0.80)* |
| **Depression** | | | | |
| Minimal (1–10) | Ref | Ref | Ref | Ref |
| Mild/Moderate (>10) | 1.77 (1.06-2.97) | 1.76 (1.04-3.00)* | 1.18(0.43-3.26) | 0.96 (0.30-3.06) |

aCOR: Crude Odds Ration; bCI: Confidence Interval; cAOR: Adjusted Odds Ratios.

*significant *Refernce for underweight and overweight/obese is normal.*

respectively [33]. In Southeast Asia, adolescent thinness was reported at 15.0%, while the Americas reported 2.5% [34]. Indonesia's prevalence of underweight is 16% with 11% overweight, and India shows a striking disparity with 47% underweight and 8.6% overweight/obese [35,36]. These differences are likely attributable to factors such as education levels, food security, hygiene, family size, gender, and cultural practices.

Our study highlighted gender as a significant determinant, with girls showing 43% lower odds of being underweight compared to boys. This finding aligns with existing research in Pakistan and other regions, where boys are more likely to be underweight [37]. The gender disparity may be due to biological differences, such as hormonal changes that promote fat deposition in females, and reduced physical activity among girls after puberty [38,39]. Gender-specific interventions are essential to address these nutritional disparities effectively.

Working status also influenced underweight status as adolescents working 0–3 hours daily had 1.6 times higher odds of being underweight compared to non-working peers. The association can be attributed to poor families, financial instability linked to low wages in rural areas leading to food insecurity [40]. Evidence from a case–control study in

Indonesia further demonstrates that nutritional outcomes among school-aged children are shaped by a constellation of socioeconomic, behavioral, and family-level characteristics. Although such household and family factors were not directly measured in the present study, these findings provide important contextual support for interpreting adolescent work as a marker of broader structural disadvantage contributing to undernutrition. Overall, the relationship between work hours and nutritional status underscores the need for socioeconomic interventions targeting rural adolescents [41].Adolescents who snacked between meals had 36% lower odds of being underweight but were twice as likely to be overweight. This finding mirrors literature associating consistent snacking with high caloric intake. However, calorie-dense snacks high in fats and sugars can lead to overweight and obesity. An analysis of NHANES data (2005–2016) found that overweight adolescents consumed an average of 1.85 snacks per day at 305.41 calories each, while obese adolescents consumed 1.97 snacks per day at 339.60 calories each [42]. These findings underscore the dual impact of snacking on undernutrition and overweight, highlighting the need for nutritional education on healthy snacking choices.

Hygiene practices were significantly linked to nutritional outcomes. Adolescents who did not wash their hands with soap had 1.47 times higher odds of being underweight. This aligns with studies showing that good hygiene practices reduce the likelihood of underweight children by decreasing exposure to pathogens that cause infections impairing nutrient absorption [43]. Poor handwashing practices, exacerbated by limited access to soap and sanitation in low-income settings, highlight the need for public health campaigns focused on hygiene to reduce undernutrition risks.

Family size was inversely related to overweight/obesity, with adolescents from larger families (more than five members) having 56% lower odds of being overweight/obese. Studies show that each additional sibling reduces the likelihood of obesity by 2.6 percentage points due to shared meals and active engagement [44]. While larger family sizes may be linked to healthier dietary habits, it's essential to understand the broader socioeconomic and cultural context to avoid oversimplification.

Lastly, depression was associated with higher odds of being underweight, with those experiencing mild to moderate depression having 1.7 times higher odds. This aligns with research indicating that depressive symptoms can lead to weight loss due to changes in appetite and eating patterns [45]. Addressing mental health is crucial for comprehensive adolescent health strategies, as untreated depression can exacerbate nutritional challenges.

While our study provides important insights into factors associated with adolescent malnutrition in rural Pakistan, it has several strengths and limitations. The study employed multinomial logistic regression, allowing simultaneous analysis of underweight and overweight/obese outcomes without inflating type I error. Inclusion of both in-school and out-of-school adolescents enhanced the representativeness of the sample, while standardized anthropometric measurements and validated, pilot-tested tools ensured reliable and consistent data collection. Despite these strengths, cross-sectional design limits the ability to establish causal relationships between risk factors and nutritional outcomes. Certain exposures, including diet, snacking, sleep, and hygiene, were self-reported and may be affected by recall or reporting bias. Additionally, some variables approached statistical significance and should be interpreted cautiously. Overall, while the study offers valuable insights, these limitations should be considered when interpreting the findings.

## Conclusion

This study underscores the importance of targeted, gender-sensitive interventions to address both undernutrition and overweight/obesity among adolescents. Key areas include improving hygiene, promoting healthy eating, and integrating mental health support into nutrition programs. Educational campaigns and improved access to clean water, sanitation, and diverse diets are essential. A comprehensive, multisectoral approach is needed to reduce DBM and improve adolescent health and nutrition.Given these limitations, future research should employ longitudinal and mixed-methods designs to more clearly establish causal pathways, validate self-reported behaviors, and explore the contextual, behavioral, and environmental factors underlying adolescent malnutrition that could not be fully captured in this cross-sectional study.

## Supporting information

**S1 Data. Anonymized dataset used for the analysis of the double burden of malnutrition among adolescents aged 10–19 years in rural Pakistan.**
(XLS)

## Acknowledgments

We gratefully acknowledge Sir Iqbal Azam, Assistant Professor in the Department of Community Health Sciences, for his invaluable support and thoughtful guidance throughout this work.

## Author contributions

**Conceptualization:** Muzna Hashmi, Zahra Ali Padhani, Ayesha Zahid Khan, Jai K Das.

**Data curation:** Muzna Hashmi, Mushtaque Mirani, Rasool Bux, Imran Ahmed Chauhadry.

**Formal analysis:** Muzna Hashmi, Naureen Rehman, Arjumand Rizvi, Bhavita Kumari, Rasool Bux, Jai K Das.

**Investigation:** Mushtaque Mirani.

**Methodology:** Muzna Hashmi, Naureen Rehman, Arjumand Rizvi, Jai K Das.

**Project administration:** Muhammad Khan, Sana khatoon.

**Software:** Muzna Hashmi.

**Supervision:** Jai K Das.

**Validation:** Muzna Hashmi.

**Visualization:** Muzna Hashmi.

**Writing – original draft:** Muzna Hashmi, Naureen Rehman.

**Writing – review & editing:** Muzna Hashmi, Jai K Das.

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
