## [Decision Letter · Decision Letter 0]

8 Dec 2025

PGPH-D-25-03040

Double burden of malnutrition and its associated factors among adolescents aged (10-19) years in a rural district of Pakistan.

Dear Dr. Das,

Thank you for submitting your manuscript to PLOS Global Public Health. After careful consideration, we feel that it has merit but does not fully meet PLOS Global Public Health’s publication criteria as it currently stands. Therefore, we invite you to submit a revised version of the manuscript that addresses the points raised during the review process.

We look forward to receiving your revised manuscript.

Kind regards,

Muhammad Iqhrammullah, Ph.D

Academic Editor

Journal Requirements:

Additional Editor Comments (if provided):

The double burden of malnutrition does not mean pooling underweight and overweight/obesity into one category. Instead, DBM is demonstrated when both conditions exist in the same population. Therefore, underweight and overweight/obese should be presented as separate categories, and their coexistence in the population indicates the presence of DBM. Avoid pooling the prevalence!

Reviewers' comments:

Reviewer's Responses to Questions

**Comments to the Author**

1. Does this manuscript meet PLOS Global Public Health’s publication criteria? Is the manuscript technically sound, and do the data support the conclusions? The manuscript must describe methodologically and ethically rigorous research with conclusions that are appropriately drawn based on the data presented.

Reviewer #1: Partly

Reviewer #2: Yes

2. Has the statistical analysis been performed appropriately and rigorously?

Reviewer #1: No

Reviewer #2: No

3. Have the authors made all data underlying the findings in their manuscript fully available (please refer to the Data Availability Statement at the start of the manuscript PDF file)?

Reviewer #1: No

Reviewer #2: No

4. Is the manuscript presented in an intelligible fashion and written in standard English?

Reviewer #1: Yes

Reviewer #2: Yes

5. Review Comments to the Author

Reviewer #1: 1. Ensure consistency in reporting numerical results. In the abstract, the percentage for DBM is stated as 22.5%, but in the discussion section, it is written as 22.58%. Please change it so the percentage is consistently 22.5%.

2. Pay attention to the formatting of the letter 'p' denoting p-value. This letter should be italicized.

3. (Reference #32): Please check the references. This literature source is not cited in the manuscript text.

4. Conduct a thorough proofread of the entire manuscript to avoid grammatical errors and typos, such as the term "over nutrition", which should be written as one word "overnutrition".

5. (Tables): Where is Table 4? In the manuscript, after Table 3, it goes directly to Table 5. Is this a numbering error, and should Table 5 actually be numbered as Table 4?

6. (Results, line 211): There is an inconsistency in data presentation. In the abstract, the value for underweight is written as “95% CI: 15.0–20.0”. However, in the results section, it is written as “95% CI: 0.15–0.20”. Please correct this notation.

7. (Results): Strengthen the explanation of how the selection process proceeded from 1,304 households to only 1,185 adolescents being deemed eligible for selection

8. Please add an explanation regarding missing data in this study. Was there any missing data? How much? And how was it handled?

9. (Discussion): Strengthen the discussion related to Socio-Economic Factors (Working Status). Please check and cite the following article: https://doi.org/10.52225/narra.v5i1.2049

10. Strengthen the explanation of the limitations faced in this study, as the limitations currently discussed are still insufficient and overly simplistic.

Reviewer #2: Background

The manuscript addresses an important and timely public health issue—the double burden of malnutrition (DBM) among adolescents in rural Pakistan. The introduction is well-structured, provides strong global and regional context, and clearly highlights the knowledge gap in rural Pakistani settings. The justification for focusing on the TMK district is also well explained.

Methodology

The methodology is generally sound and clearly presented. The use of a representative multistage cluster sampling approach strengthens the validity of the findings. The sample size is adequate, and the use of standardized WHO anthropometric criteria is appropriate. The statistical analysis using multinomial logistic regression with survey design adjustments is rigorous and suitable for examining the determinants of DBM. Ethical considerations should be clearly stated in the full manuscript, including ethics approval, consent procedures, and data confidentiality.

Results

The results are presented clearly and are supported by the statistical analysis. The prevalence figures for underweight and overweight/obesity are consistent with existing regional trends. The identification of both risk and protective factors is useful for programmatic planning. Confidence intervals are appropriately reported. Some variables approach marginal significance, and this should be acknowledged in the discussion.

Discussion

The discussion appropriately interprets the findings in the context of existing literature. The manuscript highlights how lifestyle, gender, hygiene, mental health, and socioeconomic conditions shape adolescent nutritional outcomes. However, the discussion could benefit from deeper reflection on causal pathways and potential confounding factors. Strengths and limitations of the study—such as reliance on cross-sectional data and possible reporting bias—should also be elaborated more fully.

Conclusion

The conclusions are consistent with the results and emphasize the need for integrated interventions addressing hygiene, mental health, dietary practices, and gender-specific needs. The recommendations are relevant and aligned with public health priorities in low-resource settings.

Overall Comment

This is a well-written and coherent manuscript that meets publication standards in most areas. Minor improvements in clarity, formatting, and elaboration of methodological limitations will strengthen the manuscript further.

6. PLOS authors have the option to publish the peer review history of their article (what does this mean?). If published, this will include your full peer review and any attached files.

**Do you want your identity to be public for this peer review?** For information about this choice, including consent withdrawal, please see our Privacy Policy.

Reviewer #1: No

Reviewer #2: No

Figure Resubmissions:

---

## [Decision Letter · Decision Letter 1]

15 Jan 2026

PGPH-D-25-03040R1

Double burden of malnutrition and its associated factors among adolescents aged (10-19) years in a rural district of Pakistan.

Dear Dr. Das,

Thank you for submitting your manuscript to PLOS Global Public Health. After careful consideration, we feel that it has merit but does not fully meet PLOS Global Public Health’s publication criteria as it currently stands. Therefore, we invite you to submit a revised version of the manuscript that addresses the points raised during the review process.

We look forward to receiving your revised manuscript.

Kind regards,

Muhammad Iqhrammullah, Ph.D

Academic Editor

Journal Requirements:

Additional Editor Comments (if provided):

Reviewers' comments:

Reviewer's Responses to Questions

**Comments to the Author**

1. If the authors have adequately addressed your comments raised in a previous round of review and you feel that this manuscript is now acceptable for publication, you may indicate that here to bypass the “Comments to the Author” section, enter your conflict of interest statement in the “Confidential to Editor” section, and submit your "Accept" recommendation.

Reviewer #1: All comments have been addressed

Reviewer #2: All comments have been addressed

2. Does this manuscript meet PLOS Global Public Health’s publication criteria? Is the manuscript technically sound, and do the data support the conclusions? The manuscript must describe methodologically and ethically rigorous research with conclusions that are appropriately drawn based on the data presented.

Reviewer #1: Yes

Reviewer #2: Yes

3. Has the statistical analysis been performed appropriately and rigorously?

Reviewer #1: Yes

Reviewer #2: Yes

4. Have the authors made all data underlying the findings in their manuscript fully available (please refer to the Data Availability Statement at the start of the manuscript PDF file)?

Reviewer #1: Yes

Reviewer #2: Yes

5. Is the manuscript presented in an intelligible fashion and written in standard English?

Reviewer #1: Yes

Reviewer #2: Yes

6. Review Comments to the Author

Reviewer #1: 1. (Methods): Please clarify which prevalence estimate was used (30.7% or 20.1%) and how they were combined. Was the higher prevalence (30.7%) used to ensure an adequate sample size, or was the 20.1% figure used?

2. (Methods): Please complete the exclusion criteria in the Methods section. Additional criteria such as pregnancy, psychiatric conditions, and chronic diseases were only mentioned in the Results section but were not stated earlier in the Methods.

3. (Conclusion): Please further strengthen the conclusion by adding a sentence that explicitly recommends future research, related to the study's limitations that still require deeper exploration.

4. Overall, this manuscript is well-structured and addresses an important public health issue. The authors have been responsive to the previous reviewers' comments, and the revisions have significantly improved the manuscript. The study is methodologically sound, and the findings provide valuable insights into the double burden of malnutrition among adolescents in rural Pakistan. Therefore, I recommend acceptance with minor revisions to strengthen the manuscript further.

Reviewer #2: This manuscript presents a methodologically sound and well-executed study on the double burden of malnutrition among adolescents in rural Pakistan. The study design, sampling strategy, anthropometric measurements based on WHO standards, and use of multinomial logistic regression are appropriate and rigorously applied. The statistical analyses support the conclusions drawn.

The manuscript is clearly written in standard English, and the discussion appropriately situates the findings within existing literature while acknowledging study limitations. Data availability is adequately addressed in accordance with PLOS policy, and no ethical or publication concerns were identified.

Overall, the manuscript meets the publication criteria of PLOS Global Public Health and is suitable for publication.

7. PLOS authors have the option to publish the peer review history of their article (what does this mean?). If published, this will include your full peer review and any attached files.

**Do you want your identity to be public for this peer review?** For information about this choice, including consent withdrawal, please see our Privacy Policy.

Reviewer #1: No

Reviewer #2: No

 Figure Resubmissions:

---

## [Decision Letter · Decision Letter 2]

29 Mar 2026

PGPH-D-25-03040R2

Double burden of malnutrition and its associated factors among adolescents aged (10-19) years in a rural district of Pakistan.

Dear Dr. Das,

Thank you for submitting your manuscript to PLOS Global Public Health. After careful consideration, we feel that it has merit but does not fully meet PLOS Global Public Health’s publication criteria as it currently stands. Therefore, we invite you to submit a revised version of the manuscript that addresses the points raised during the review process.

We look forward to receiving your revised manuscript.

Kind regards,

Helen Howard

Staff Editor

**Journal Requirements:**

**Additional Editor Comments (if provided):**

Reviewers' comments:

Reviewer's Responses to Questions

**Comments to the Author**

1. If the authors have adequately addressed your comments raised in a previous round of review and you feel that this manuscript is now acceptable for publication, you may indicate that here to bypass the “Comments to the Author” section, enter your conflict of interest statement in the “Confidential to Editor” section, and submit your "Accept" recommendation.

Reviewer #1: All comments have been addressed

Reviewer #2: All comments have been addressed

Reviewer #3: All comments have been addressed

2. Does this manuscript meet PLOS Global Public Health’s publication criteria? Is the manuscript technically sound, and do the data support the conclusions? The manuscript must describe methodologically and ethically rigorous research with conclusions that are appropriately drawn based on the data presented.

Reviewer #1: Yes

Reviewer #2: Yes

Reviewer #3: Yes

3. Has the statistical analysis been performed appropriately and rigorously?

Reviewer #1: Yes

Reviewer #2: Yes

Reviewer #3: Yes

4. Have the authors made all data underlying the findings in their manuscript fully available (please refer to the Data Availability Statement at the start of the manuscript PDF file)?

Reviewer #1: Yes

Reviewer #2: Yes

Reviewer #3: Yes

5. Is the manuscript presented in an intelligible fashion and written in standard English?

Reviewer #1: Yes

Reviewer #2: Yes

Reviewer #3: Yes

6. Review Comments to the Author

**Reviewer #1:** 1. (Abstract): Please ensure consistency in the use of terminology. In the results section of the abstract, the term "lack of handwashing" is used, but in the tables/analysis, the variable is referred to as "no soap available/handwash supplies." Please standardize this to avoid confusing the reader.

2. (Results): Regarding the number of respondents, page 14 (lines 204-205) states that 1,185 households had an eligible adolescent. In the following paragraph (line 209), the analysis was conducted on 1,159 adolescents. What happened to the other 26 adolescents? Please explain this in detail. Figure 1 also only shows 1,185 eligible, and then directly 1,159. There is no "excluded" box with the reasons.

3. Overall, this article is already very strong after going through several previous revisions. The authors have also addressed the reviewers' comments very well. Perhaps with a few minor improvements in some sections, as I have mentioned above, this article is suitable for publication.

**Reviewer #2:** Thank you for the revisions. The authors have adequately addressed the reviewer comments and editorial suggestions. The manuscript has improved substantially and is now suitable for publication. I have no further comments.

**Reviewer #3:**The manuscript focuses on a public health matter of importance, the double burden of adolescent undernutrition and overnutrition in rural Pakistan. The author's methodological approach to studying the double burden of malnutrition among the adolescent population and its determinants is comprehensively explained, sound, with a well-designed study and robust statistical analysis. The articulation of the study findings is in accordance with the methods and analysis, and the discussion contributes by bringing perspectives from studies conducted elsewhere.

The authors have fully and satisfactorily addressed the comments from the second review as indicated in the response sheet.

7. PLOS authors have the option to publish the peer review history of their article (what does this mean?). If published, this will include your full peer review and any attached files.

**Do you want your identity to be public for this peer review?** For information about this choice, including consent withdrawal, please see our Privacy Policy.

Reviewer #1: No

Reviewer #2: No

Reviewer #3: No

**Figure Resubmissions:**While revising your submission, we strongly recommend that you use PLOS’s NAAS tool (https://ngplosjournals.pagemajik.ai/artanalysis) to test your figure files. NAAS can convert your figure files to the TIFF file type and meet basic requirements (such as print size, resolution), or provide you with a report on issues that do not meet our requirements and that NAAS cannot fix.

---

## [Editor Report · Decision Letter 3]

13 Apr 2026

PGPH-D-25-03040R3

Double burden of malnutrition and its associated factors among adolescents aged (10-19) years in a rural district of Pakistan.

Dear Dr. Das,

Thank you for submitting your manuscript to PLOS Global Public Health. After careful consideration, we feel that it has merit but does not fully meet PLOS Global Public Health’s publication criteria as it currently stands. Therefore, we invite you to submit a revised version of the manuscript that addresses the points raised during the review process.

We look forward to receiving your revised manuscript.

Kind regards,

Helen Howard

Staff Editor

**Journal Requirements:**

**Additional Editor Comments:**

- We note that Figure 1 says "Not competent 25". Should this be 26? Please check and correct as necessary.

**Reviewers' comments:**

 **Figure Resubmissions:**While revising your submission, we strongly recommend that you use PLOS’s NAAS tool (https://ngplosjournals.pagemajik.ai/artanalysis) to test your figure files. NAAS can convert your figure files to the TIFF file type and meet basic requirements (such as print size, resolution), or provide you with a report on issues that do not meet our requirements and that NAAS cannot fix.

---

## [Editor Report · Decision Letter 4]

27 Apr 2026

Double burden of malnutrition and its associated factors among adolescents aged (10-19) years in a rural district of Pakistan.

PGPH-D-25-03040R4

Dear Dr. Das,

We are pleased to inform you that your manuscript 'Double burden of malnutrition and its associated factors among adolescents aged (10-19) years in a rural district of Pakistan.' has been provisionally accepted for publication in PLOS Global Public Health.

Best regards,

Julia Robinson

Executive Editor